# Hyperspectral Compute-In-Memory:
# An Opto-Electronic Computing Architecture
# Enabling Compute Density Beyond PetaOPS/mm$^2$

**Myoung-Gyun Suh**[*]**, Byoung Jun Park, Mostafa Honari Latifpour, Yoshihisa Yamamoto**
Physics & Informatics Laboratories, NTT Research, Inc., Sunnyvale, CA 94085, USA
[*]`email:myoung-gyun.suh@ntt-research.com`

## Abstract

We present a hyperspectral compute-in-memory architecture that utilizes both frequency and spatial dimensions for single-shot matrix-matrix multiplication. This approach offers exceptional parallelism, scalability, programmability, and efficient chip area utilization, potentially enabling a compute density exceeding PetaOPS/mm$^2$. The architecture demonstrates potential for energy-efficient, three-dimensional opto-electronic computing in future data center applications.

Recent advancements in artificial intelligence (AI) have revolutionized various industries[1]. As AI models grow exponentially in size, traditional electronic systems are struggling to keep up due to inherent scaling limitations. This has necessitated the deployment of extensive networks of disaggregated electronic chips dedicated to individual computational tasks, as seen with modern GPT models that require thousands of GPUs. As a result, optical technologies have become increasingly significant in data centers, enhancing data transfer alongside electrical systems and catalyzing the evolution of data centers into hybrid optical/electrical computing environments. Optical interconnect technologies are advancing to more closely integrate with electronic chips, driven by the demand for higher bandwidth capacities. Challenges in increasing serializer/deserializer (SerDes) speeds have spurred strategies like space and frequency multiplexing to expand bandwidth. Moreover, researchers are exploring methods to reduce power consumption within single electronic chips, especially in traditional von Neumann architectures, leading to the exploration of compute-in-memory (or in-memory computing) architectures[2]. By integrating non-volatile memory components within processors, these systems avoid data transfer bottlenecks between memory and processing units, thereby enhancing data efficiency, reducing power usage, and enabling highly parallel computations.

As data centers transition to hybrid opto-electronic platforms, it becomes pertinent to consider if optics could handle computational tasks typically assigned to electronics. Since linear operations are particularly suited for optical computing among various computational tasks, there is renewed interest in utilizing optics for energy-efficient matrix-vector multiplication (MVM)[3; 4]. This has led to the proposal and demonstration of numerous optical MVM systems in recent years[5; 6; 7; 8; 9]. In this context, three-dimensional (3D) optical systems employing scalable free-space optics are particularly promising[6; 7; 8; 9; 10; 11]. Yet, most systems to date primarily utilize space multiplexing, with the frequency dimension remaining underexplored. Our work introduces a hyperspectral compute-in-memory architecture that merges space and frequency multiplexing, boosting computational efficiency and throughput[12] (See Figure 1a). This architecture optimizes energy use and reduces data movement via in-memory computing. Our system processes optical signals through a two-dimensional (2D) spatial light modulator (SLM)[13; 14; 15], functioning as programmable optical memory, enabling parallel operations across spatial dimensions. This setup utilizes optics to efficiently handle parallel data processing, while electronics enhance programmability. Considering

Submitted to the Second Workshop on Machine Learning with New Compute Paradigms at NeurIPS (MLNCP 2024). Do not distribute.

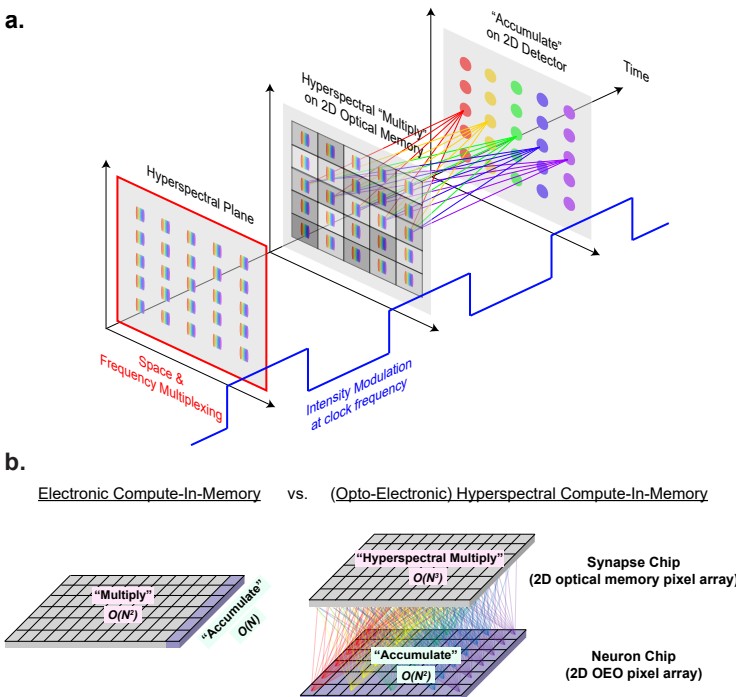

Figure 1: (a) Hyperspectral Compute-In-Memory (CIM) architecture enhances computational throughput by integrating space and frequency multiplexing at each computational clock cycle. (b) Unlike its electronic counterparts, the Opto-Electronic Hyperspectral CIM architecture eliminates the need for physical wiring in MAC operations, enabling a 3D architectural design. By dividing the "multiply" and "accumulate" operations across two distinct chips (a synapse chip and a neuron chip), the architecture optimizes chip area utilization and is capable of achieving a compute density exceeding PetaOPS/mm$^2$.

the lower density limitations of space multiplexing compared to electronic systems, our architecture additionally integrates frequency multiplexing with optical frequency combs (OFCs)(16; 17), drawing inspiration from hyperspectral imaging(18) and advanced optical fiber communications(19).

In our proof-of-concept experiments, we manipulate 2D optical input data for single-shot matrix-matrix multiplication (MMM), where each SLM pixel encodes a matrix weight across multiple wavelengths. This method allows batch processing of matrix-vector multiplication using wavelength-division multiplexing. We conducted numerous MMM tests, and the results confirmed theoretical predictions, including the multiplication of the NTT logo with the identity matrix, as shown in Figure 2d. Although hyperspectral imaging usually involves 3D data both in input and output, our computing system maintains 2D inputs and outputs, utilizing the third dimension internally. This strategy transforms the "curse of dimensionality" into a computational asset.

Figure 2a illustrates the experimental setup for demonstrating the hyperspectral compute-in-memory architecture. The input source is a fiber optical frequency comb (OFC) in the C-band, featuring a 250 MHz pulse repetition rate and is coarsely filtered using line-by-line waveshaping(14) as shown in Figure 2b. The optical source, with an average power of around 1 mW, is then introduced into the system. The coarsely filtered comb lines are spatially dispersed using a grating, expanded vertically by a cylindrical lens, and then focused onto SLM 1, where the first matrix is encoded. The comb lines are then recombined and expanded horizontally by another cylindrical lens before being focused onto SLM 2 to encode the second matrix. After another vertical fanning-in by a cylindrical lens, the comb lines are sorted vertically by color via a grating to complete the hyperspectral multiply-accumulate operation. A linear polarizer enables the phase-only SLM to modulate intensity, and system non-uniformity is calibrated by adjusting the SLM pixel phases.

To demonstrate the hyperspectral operation, we conducted MMM tests with a hyperspectral factor of 5, encoding each SLM pixel with a matrix weight across five comb lines (see Figure 3a). Minor

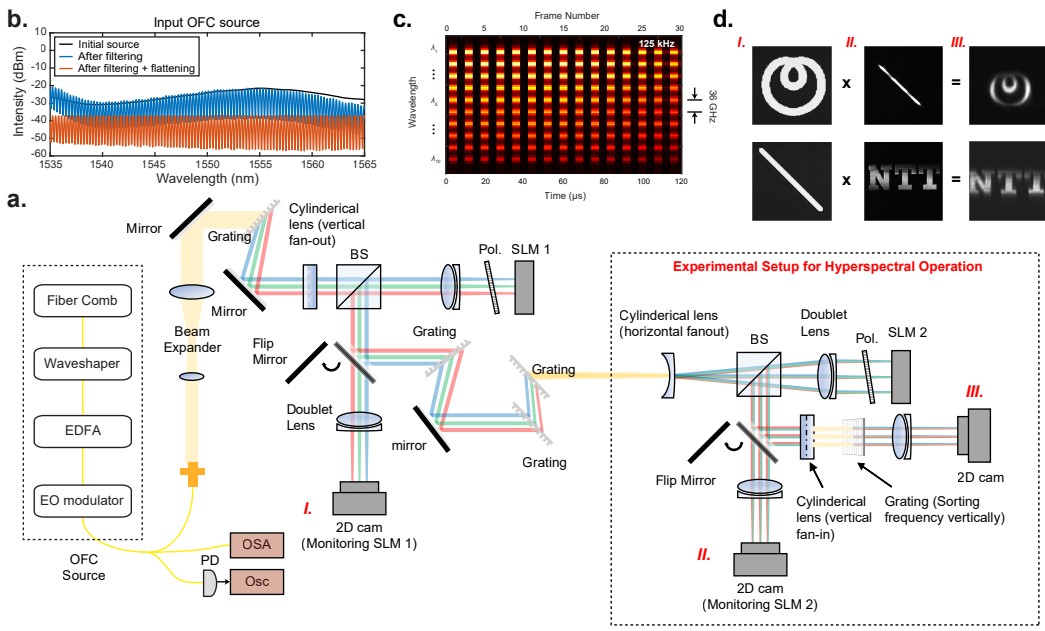

Figure 2: (a) Experimental setup for the open-loop hyperspectral multiply-accumulate (MAC) operation, enabling single-shot matrix-matrix multiplication (MMM). Matrices are projected onto SLM 1 and SLM 2, with the resulting matrix captured by a 2D camera. (b) Displays typical optical spectra of the input OFC source, shown before and after spectral filtering and flattening. (c) Illustrates the time evolution of line-scan camera images at a frame rate of 250 kHz, which depicts the intensity modulation of the input OFC source with 10 frequency components spaced by 36 GHz. The modulation rate of the intensity is 125 kHz. (d) Displays the encoding of the NTT logo and an identity-like matrix (I and II), each approximately 300 by 300 in size, resulting in an output matrix that displays the NTT logo (III).

adjustments to our system allowed for a potential increase in the hyperspectral factor to 10 or higher. We evaluated the computational accuracy by analyzing the error distribution for each possible MAC value. The matrices were encoded using non-negative weights with 4 bits. We performed 400 measurements for each MAC value, ranging from 0 to 150 (see Figure 3b). As the target MAC values increased, the standard deviation of the error grew until reaching a saturation point. The relative error, defined as the absolute difference between the measured and target MAC values divided by the target MAC value, showed a standard deviation decreasing to below 5 percent as the target MAC value increased. These errors likely arose from intensity fluctuations in the OFC source, crosstalk between adjacent pixels, and optical alignment errors. We anticipate that the standard deviation of the relative error will stabilize at a similar level even when the system scales up in matrix size. Notably, noise up to a certain threshold may not significantly affect computational outcomes in many AI tasks, as confirmed by analyzing MNIST data classification under various noise conditions.

The system currently operates in an open-loop configuration, encoding the input matrix and independently reading out MAC results using standard digital electronics. Fast external modulation and readout are vital for high-throughput computation in such setups. Conversely, in a closed-loop configuration with nonlinear operations, the system efficiently solves optimization problems without the need for rapid external modulation and readout. Most computations here are analog, with only the initial input and final output digitally managed. To enable rapid, pixel-by-pixel parallel modulation in the closed-loop system, a novel 2D opto-electronic "neuron" array is essential. This array connects each photodetector pixel directly to its corresponding modulator (or light emitter) pixel via through-silicon-via (TSV), reducing delays and energy consumption by avoiding the inefficiencies of connecting a camera to an SLM via a serial bus. Such an array would enable seamless parallel processing.

In the near term, we aim to operate our MMM system in closed-loop mode (refer to Figure 4b), primarily for its simplicity. This configuration requires just one hyperspectral MAC module, and it

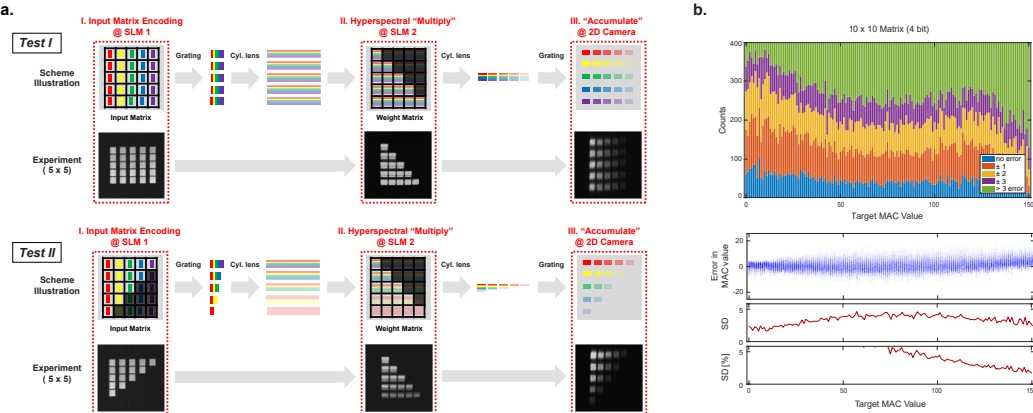

Figure 3: (a) Illustrations of Matrix-Matrix Multiplication (MMM) through hyperspectral operation are presented alongside images from two test experiments. The theoretical diagrams closely match the experimental results. Test I features the multiplication of an all-ones matrix with a lower triangular matrix, while Test II illustrates the multiplication of two triangular matrices. To simplify the demonstration, examples with a hyperspectral factor of 5 are used. (b) The error distribution for each possible MAC value is displayed. For each MAC value, 400 MAC operations are conducted for analysis. The data originates from a 10 x 10 matrix with a hyperspectral factor of 10. The absolute error, the standard deviation (SD) of the error distribution, and the error percentage are further detailed in the lower panels.

removes the need for parallel modulation and readout through an external electronic interface. In this setup, only one external intensity modulation at the computational clock frequency is necessary to generate the input optical pulse stream. To determine the total power consumption of this system, we calculated the power required for each pixel during $N_b$-bit precision MAC operations, using actual parameters and factoring in the significant fixed energy costs from our current experimental setup. Given the hyperspectral factor $H$ for multiplying matrices of sizes $(H \times K)$ and $(K \times K)$, the system executes approximately $(H \times K \times K)$ MAC operations per single clock cycle, and the formula for total power consumption is as follows:

$$P_{H \times K \times K}^{\text{(closed-loop MMM)}} \approx P_{mod} + \left\{ P_{SLM} + (H \times K) \times \left[ \frac{2^{N_b} I_{th}}{\eta_L \eta_o' \eta_{PD}} + P_{TIA} \right] \right\}. \quad (1)$$

Here, $N_b$ represents the effective bit precision, $I_{th}$ is the threshold current for detection in the photodetector, $\eta_{PD}$ denotes the photodetector responsivity, $\eta_L$ refers to the laser wall-plug efficiency, $\eta_o'$ is the efficiency of optical power utilization, and $P_{mod}$, $P_{SLM}$, and $P_{TIA}$ are the respective power consumptions for the optical modulator, the spatial light modulator (SLM), and the transimpedance amplifier.

With improved alignment and wider spectral bandwidth, the closed-loop system is expected to reach 100 peta operations per second (PetaOPS), with $H$ = 100, $K$ = 1000, and a 1 GHz clock frequency, and an anticipated efficiency close to 2 W/PetaOPS (as shown in Figure 4b and Scenario 2 of Table I). The 'hyperspectral factor' mitigates the need for extensive physical scaling. For instance, with a hyperspectral factor of 400 and maintaining the same clock speed, only a 500-by-500 matrix (i.e., $K$ = 500) is required to achieve 100 PetaOPS. Further scaling in the space and frequency dimensions could push the system beyond ExaOPS while keeping the power efficiency around 2 W/PetaOPS. A multi-layered ($L$-layer) open-loop hyperspectral system (outlined in Figure 4a and Scenario 3 of Table I) is expected to demonstrate comparable power efficiency, provided that the number of layers is sufficient to effectively offset the energy overhead from input electro-optic (EO) and output opto-electronic (OE) conversions. While direct comparisons of power consumption between mature digital electronic computing technologies and nascent optical computing lab demonstrations are challenging, our projections indicate a considerable boost in efficiency compared to state-of-the-art electronic GPUs.

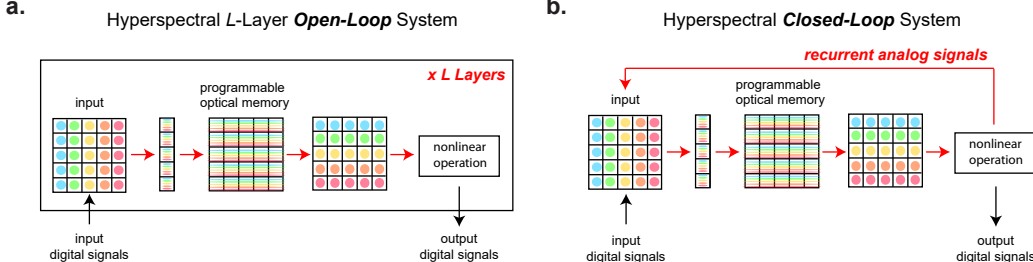

Figure 4: (a) An open-loop system featuring $L$ cascaded layers of hyperspectral Multiply-Accumulate modules positioned between the input and output digital electronic interfaces. (b) A closed-loop system functions as a physical solver for optimization problems, where optical or electrical analog signals circulate within the loop and stabilize at a steady-state solution. Note: Various optical frequencies are represented by different colors. While the analog signal pathways, marked by red arrows, support parallel data transmission, a single line is depicted for clarity.

Table 1: **Estimated System Performance**

| | Current (open-loop) | Scenario 1 (closed-loop) | Scenario 2 (closed-loop) | Scenario 3 (open-loop) |
|---|---|---|---|---|
| Number of Layers ($L$) | 1 | 1 | 1 | 50 |
| Hyperspectral Factor ($H$) | 1 (10) | 30 | 100 | 100 |
| Input Matrix Size (H $\times$ $K_1$) | $1 \times 64$ | $30 \times 300$ | $100 \times 1000$ | $100 \times 1000$ |
| Weight Matrix Size ($K_1 \times K_2$) | $64 \times 128$ | $300 \times 300$ | $1000 \times 1000$ | $1000 \times 1000$ |
| Clock Frequency | 250 MHz$^\dagger$ | 1 GHz | 1 GHz | 1 GHz |
| Computational Throughput | 2.048 TOPS (20.48 TOPS) | 2.7 PetaOPS | 100 PetaOPS | 5 ExaOPS |
| Total Power Consumption$^{\dagger\dagger}$ | 11.9 W (32.2 W) | 27.7 W | 206 W | 12.6 kW |
| Power Efficiency | 5.8 W/TOPS (1.57 W/TOPS) | 10.26 W/PetaOPS | 2.06 W/PetaOPS | 2.52 W/PetaOPS |

$^\dagger$ We assume an external modulation and readout speed of 250 MHz.
$^{\dagger\dagger}$ Details of the power consumption estimation are discussed in Reference 12.

Our hyperspectral compute-in-memory architecture operates as a 3D opto-electronic computing system, processing 2D optical input data through a 2D optical memory "synapse" that conducts an $O(N^3)$ hyperspectral "multiply" operation. Concurrently, the 2D opto-electronic "neuron" performs $O(N^2)$ "accumulate" and nonlinear activation functions in parallel at every clock cycle, ensuring minimal latency. This architecture optimally uses chip area by directly linking the "synapse" and "neuron" chips optically, removing the need for physical wires and potentially achieving a compute density that exceeds PetaOPS/mm$^2$ (See Figure 1b). Significantly, by localizing electronic operations within each pixel during computation, this setup minimizes electronic data movement, with most data communication handled optically. This efficiency substantially offsets the costs associated with electrical-to-optical (EO) and optical-to-electrical (OE) conversions.

Our proposed hyperspectral in-memory computing system fully utilizes the dimensions of frequency, space, and time to enhance computational throughput and energy efficiency. It integrates space and frequency multiplexing using scalable SLM and OFC technologies, which are seeing rapid advancements through both industry and academic contributions. The modular nature of this design not only enables manufacturing by leveraging existing technologies and ecosystems but also encourages enhancements in individual component technologies, thereby driving overall system performance improvements. As scalability extends, incorporating optical element arrays and polarization multiplexing is envisaged, though large computational tasks are likely to be distributed across multiple small-scale optical computing modules, similar to traditional electronic systems. Integrating advanced optical components like metalenses[20], chip-integrated OFCs[21], and amplifiers[22] into a single or fewer optical elements as part of a modular assembly, suggests a trajectory towards significant system miniaturization. This advancement enables the integration of these systems into data centers as rack-mounted solutions. With ongoing improvements in component technology and the increasing importance of optics in data centers, this 3D opto-electronic computing architecture has the potential to revolutionize high-performance accelerated computing hardware in future data center applications.

- Note: Most of the experimental data and figures are from our recent paper published in Optica[12].

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
