# OpenReview forum: "Hyperspectral Compute-In-Memory: An Opto-Electronic Computing Architecture Enabling Compute Density Beyond PetaOPS/mm$^2$"
_NeurIPS.cc/2024/Workshop/MLNCP — MLNCP Poster_

### Official Review · Reviewer_KTkK · 2024-10-08
**Well written, but not entirely novel matrix-matrix multiplier using frequency and spatial multiplexing**

**Rating:** 6
**Confidence:** 3

**Review:**

The authors present a hyperspectral compute-in-memory architecture that utilizes both frequency and spatial dimensions for matrix-matrix multiplication (MMM) in order to further improve the system throughput.
The paper is well written and it is primarily a work in progress and the only reported results are about accuracy of the MMM. I am already aware of several papers proposing space and wavelength as two possible dimensions to explore to scale out the system.
See for example the below reference:
M. Honari, B. Park, Y. Kim, and M. Suh, "Matrix-Vector Multiplication using Mixed Space-Frequency Multiplexing of Optical Frequency Combs," in Optica Nonlinear Optics Topical Meeting 2023, Technical Digest Series (Optica Publishing Group, 2023), paper Tu3B.3.

So given the fact that they report preliminary work and the lack of novelty of the idea they propose, I would say that the paper is borderline acceptable, given that what they presented is correct.

---

### Decision · Program_Chairs · 2024-10-10

Accept (Poster)